# Local mechanical forces promote polarized junctional assembly and axis elongation in *Drosophila*

Jessica C Yu[1], Rodrigo Fernandez-Gonzalez[1,2,3]*

[1]Institute of Biomaterials and Biomedical Engineering, University of Toronto, Toronto, Canada; [2]Department of Cell and Systems Biology, University of Toronto, Toronto, Canada; [3]Developmental and Stem Cell Biology Program, The Hospital for Sick Children, Toronto, Canada

**Abstract** Axis elongation is a conserved process in which the head-to-tail or anterior-posterior (AP) axis of an embryo extends. In *Drosophila*, cellular rearrangements drive axis elongation. Cells exchange neighbours by converging into transient multicellular vertices which resolve through the assembly of new cell interfaces parallel to the AP axis. We found that new interfaces elongate in pulses correlated with periodic contractions of the surrounding cells. Inhibiting actomyosin contractility globally, or specifically in the cells around multicellular vertices, disrupted the rate and directionality of new interface assembly. Laser ablation indicated that new interfaces sustained greater tension than non-elongating ones. We developed a method to apply ectopic tension and found that increasing AP tension locally increased the elongation rate of new edges by more than twofold. Increasing dorsal-ventral tension resulted in vertex resolution perpendicular to the AP direction. We propose that local, periodic contractile forces polarize vertex resolution to drive *Drosophila* axis elongation.

*For correspondence: rodrigo.
fernandez.gonzalez@utoronto.ca

**Competing interests:** The authors declare that no competing interests exist.

## Introduction

Axis elongation is a conserved morphogenetic process is which the basic body plan of an animal is established. In vertebrates, axis elongation involves convergence and extension movements mediated by cell intercalation, cell migration, and oriented cell division (*Bénazéraf and Pourquié, 2013*). In *Drosophila*, axis elongation occurs in an epithelial monolayer referred to as the germband, which lengthens by more than two-fold along the anterior-posterior (AP) axis of the animal, while narrowing along the dorsal-ventral (DV) axis (*Figure 1—figure supplement 1A*). The changes in germband architecture are largely driven by cell intercalation (*Irvine and Wieschaus, 1994*).

Cell intercalation facilitates changes in tissue architecture through neighbour exchange events. In vertebrates, cell intercalation drives many developmental processes, including primitive streak formation in chick embryos (*Voiculescu et al., 2007*); gut organogenesis (*Chalmers and Slack, 2000*), neural tube closure (*Davidson and Keller, 1999*), and elongation of kidney tubules (*Lienkamp et al., 2012*) in *Xenopus*; epiboly in *Xenopus* (*Keller, 1980*) and zebrafish (*Warga and Kimmel, 1990*); convergence and extension of the mesoderm in *Xenopus* (*Wilson et al., 1989*; *Shih and Keller, 1992*), zebrafish (*Yin et al., 2008*), and mouse (*Yen et al., 2009*); and visceral endoderm migration (*Migeotte et al., 2010*; *Trichas et al., 2012*), eye lid closure (*Heller et al., 2014*), neural plate elongation (*Williams et al., 2014*), palate fusion (*Kim et al., 2015*), and limb bud elongation (*Lau et al., 2015*) in mouse.

During *Drosophila* axis elongation, cell intercalation is driven by polarized actomyosin contractility, which promotes the disassembly of interfaces separating anterior and posterior cell neighbours

**eLife digest** Tissues and organs form certain shapes that allow them to perform particular roles in the body. For example, the lungs form sacs that accommodate large volumes of air, while the skin forms a sheet to cover and protect our internal organs. One way to shape a tissue is for cells to swap places with their neighbours. During this rearrangement, the contacts between neighbouring cells break down before new contacts are formed with other cells. While the physical and molecular signals that guide the break down of cell contacts are well understood, less is known about how new contacts form.

Early in development, animal embryos establish a head-to-tail 'axis' that helps to guide where each tissue and organ will form in the body. In fruit fly embryos, the cell rearrangements that drive this process involve cells exchanging places with their neighbours by gathering around a single point. These temporary cell clusters are then organised via new cell contacts that form parallel to the head-to-tail axis.

Here, Yu and Fernandez-Gonzalez investigate the role of mechanical forces in forming new cell contacts as the head-tail axis elongates. The experiments show that disrupting the ability of the cells to generate mechanical forces inhibited the formation of new cell contacts and prevented cells from successfully swapping places. Conversely, when mechanical tension is applied at the rearrangement site, the assembly of new cell contacts happens faster. Furthermore, if the tension is applied in different orientations, new cell contacts form parallel to the direction of the mechanical force.

Yu and Fernandez-Gonzalez thus show that local mechanical forces direct the assembly of new cell contacts as the head-to-tail axis forms. These forces are most likely generated by cell contractions that appear to create mechanical tension at sites of cell rearrangement. How such physical forces are converted into molecular signals remains a question for future work.

(AP interfaces), to form multicellular vertices where four or more cells converge (*Bertet et al., 2004*; *Zallen and Wieschaus, 2004*; *Blankenship et al., 2006*). Polarized disassembly of cell contacts is also associated with cell intercalation in chick (*Rozbicki et al., 2015*), *Xenopus* (*Shindo and Wallingford, 2014*), and mouse embryos (*Williams et al., 2014*; *Lau et al., 2015*). Following contraction of AP interfaces in the *Drosophila* germband, multicellular vertices are systematically resolved through the assembly of new contacts separating dorsal and ventral cell neighbours (DV interfaces, *Figure 1—figure supplement 1B*, *Video 1*). While vertex resolution and the subsequent assembly of new cell-cell interfaces drive tissue elongation, little is known about the mechanisms that regulate these processes. Myosin turnover between phosphorylated and unphosphorylated states is important for the directionality of vertex resolution (*Kasza et al., 2014*). Computational modelling suggests that periodic contraction of the apical surface of germband cells, driven by pulsatile actomyosin networks, could promote the oriented assembly of new cell contacts (*Lan et al., 2015*). However, the role of actomyosin contractility in vertex resolution remains unclear.

In this study, we combine quantitative imaging with biophysical and pharmacological manipulations to investigate the mechanisms of vertex resolution in *Drosophila* axis elongation. We find that the assembly of new interfaces during vertex resolution occurs in pulses associated with the periodic contraction of the cells anterior and posterior to the multicellular vertex. Pulsed actomyosin contractility in the cells around the vertex is critical for the directionality and rate of assembly of the new cell interface. Local, ectopic AP tension is sufficient to accelerate the assembly of new interfaces, and local DV tension can reorient vertex resolution. Together, our results demonstrate that local, periodic actomyosin contractility directs the resolution of multicellular vertices and promotes the assembly of new cell contacts during polarized cell rearrangements in *Drosophila* germband extension.

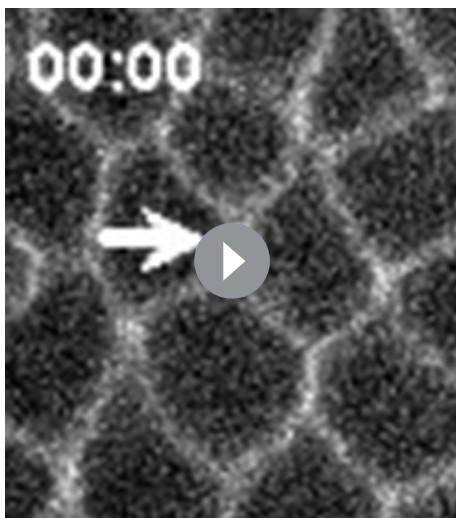

**Video 1.** Polarized cell rearrangements drive *Drosophila* axis elongation. Germband cells expressing Resille:GFP during germband extension. A stack was acquired every 10 s. Time is indicated as min:s. Anterior left, dorsal up. This video relates to *Figure 1—figure supplement 1*.

# Results

## Pulsed assembly of new junctions during germband extension

To investigate the mechanisms of vertex resolution during *Drosophila* axis elongation, we used quantitative image analysis to measure the dynamics of assembly of new DV junctions in embryos expressing Resille:GFP (*Morin et al., 2001*) to visualize cell outlines. We found that the assembly of new DV edges occurred in cycles of elongation and shortening (*Figure 1A–B*, blue line), with a period of $126 \pm 5$ s ($n = 110$ edges). On average, elongation pulses increased edge length by $772 \pm 46$ nm, while shortening pulses decreased edge length by a significantly smaller amount, $114 \pm 19$ nm ($n = 110$ edges, $p = 9.0 \times 10^{-22}$), thus resulting in net edge elongation. Germband cells undergo characteristic cycles of apical area contraction and relaxation with a period of $130 \pm 3$ s, and predominantly oriented along the AP axis of the embryo (*Fernandez-Gonzalez and Zallen, 2011*; *Sawyer et al., 2011*). To examine whether the anisotropic oscillations of germband cells were associated with the assembly of new cell junctions during vertex resolution, we compared the changes in length of the nascent DV edge to the changes in apical area of the cells immediately anterior or posterior to that DV edge (*Figure 1A–B*). In a majority of cases (143/220 cell-edge pairs, 65%), we observed a negative correlation between changes in length of the new DV junction and changes in area of the cell anterior or posterior to it (*Figure 1C*). To calculate the dominant relationship between changes in anterior/posterior cell area and new DV edge length, we quantified the correlations after shifting the edge length backward or forward in time. Reaching the maximum correlation with small time shifts would indicate in-phase oscillations, while maximum anti-correlation with small time shifts would suggest oscillations in anti-phase. We found that short time shifts of the edge length signal maximized the anti-correlation, while longer time shifts were necessary to maximize the correlation ($p = 1.74 \times 10^{-5}$, *Figure 1D–E*), further suggesting that pulses of new DV edge assembly are associated with the contraction of the anterior and posterior cells. Similar analyses demonstrated that changes in length of the new edge were predominantly positively correlated with changes in the apical area of the dorsal and ventral cells, which share the new edge (156/220 cell-edge pairs, 71%, *Figure 1—figure supplement 2*). Together, our results suggest that pulsed contractions of the cells in the immediate vicinity of a multicellular vertex may promote vertex resolution during *Drosophila* axis elongation.

## Actomyosin-induced tension is necessary and sufficient for directional vertex resolution

The cyclical changes of apical area in germband cells are driven by pulsatile networks of medial-apical non-muscle myosin II (*Rauzi et al., 2010*; *Fernandez-Gonzalez and Zallen, 2011*; *Sawyer et al., 2011*). To investigate if actomyosin contractility is necessary for vertex resolution, we injected embryos expressing E-cadherin:GFP and myosin:mCherry with the Rho-kinase inhibitor Y-27632 at 100 mM. Rho-kinase is one of the main activators of myosin (*Amano et al., 1996*; *Kimura et al., 1996*), and treatment with Y-27632 abolishes the ability of germband cells to generate mechanical force (*Fernandez-Gonzalez et al., 2009*). In Y-27632-injected embryos, germband cells displayed a rapid loss of myosin from their apical surface (*Figure 1—figure supplement 3A*), resulting in a dramatic reduction in the amplitude of apical area oscillation ($p = 1.7 \times 10^{-44}$, *Figure 1F–H*). Inhibiting actomyosin contractility affected the directionality of vertex resolution: 9/25 vertices resolved within

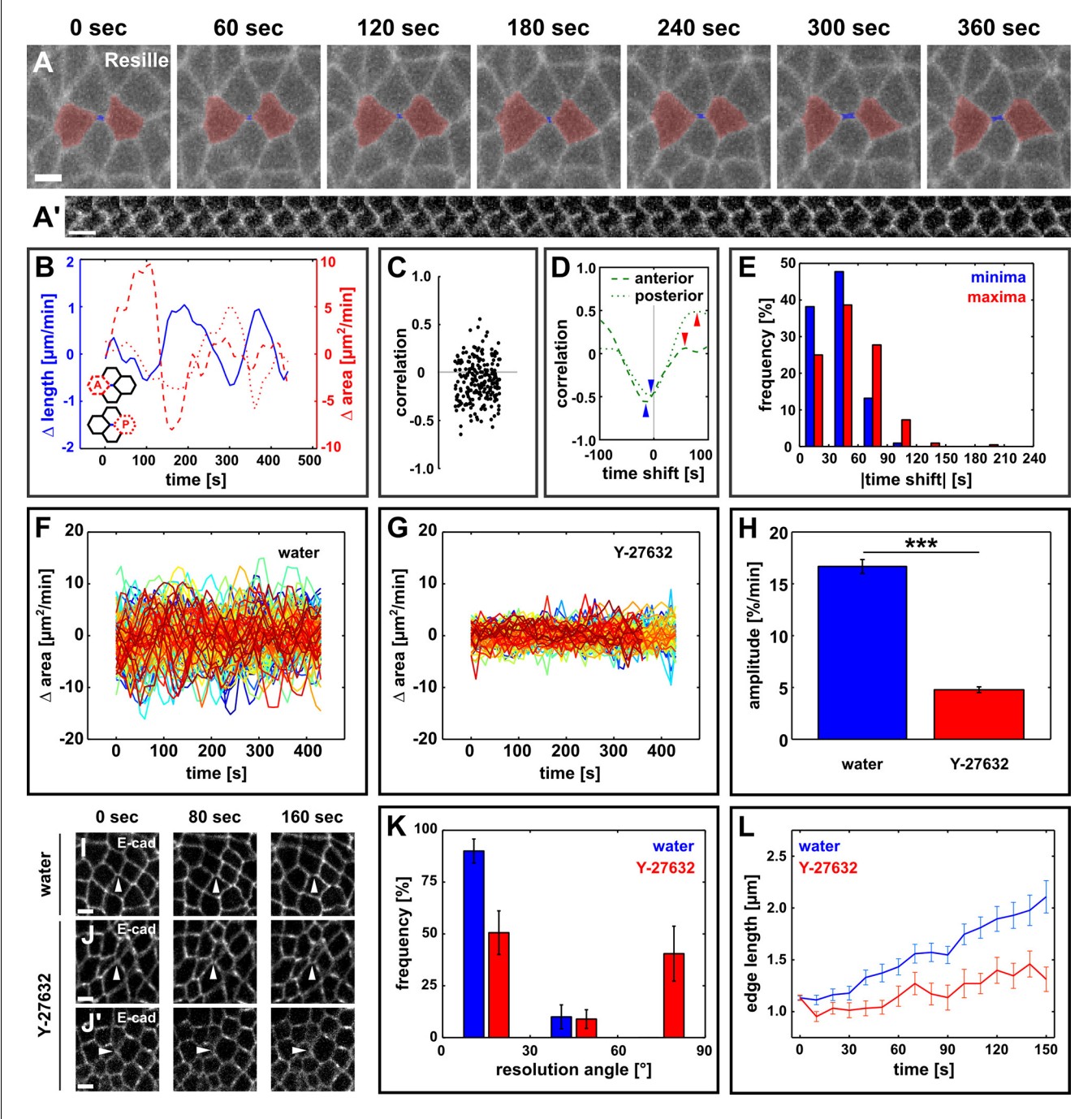

**Figure 1.** Directional assembly of new interfaces during vertex resolution is associated with pulsatile apical contractions and requires contractile activity.
(A) Vertex resolution during axis elongation in an embryo expressing Resille:GFP. Blue indicates the new DV interface, red labels the anterior and posterior cells. (A') Kymograph illustrating the elongation of the DV interface shown in (A). Scale bar, 10 s. The interface is rotated by 90° with respect to (A). Anterior down, dorsal left. (B) Rates of change for edge length (blue, solid line), anterior cell area (red, dashed line), and posterior cell area (red, dotted line) during the neighbour exchange event shown in (A). Rate of change was calculated with respect to t + 60 s. (C) Correlation coefficients between changes in edge length and changes in anterior or posterior cell area ($n$ = 220 pairs in 110 neighbour exchange events in 13 embryos). (D) Changes in correlation between edge length and anterior (dashed) or posterior (dotted) cell area during the neighbour exchange event shown in (A) when the edge length signal was shifted in time in 10-s increments. Arrowheads indicate the correlation minima (blue) or maxima (red) closest to 0-s shift. (E) Distribution of time shifts (absolute value) required to obtain the minimum (blue) and maximum (red) correlations in all 220 signal pairs shown in (C). (F, G) Rate of change in cell area in embryos injected with water (F, $n$ = 122 cells in 3 embryos) or 100 mM Y-27632 (G, $n$ = 99 cells in 3 embryos). Each line represents a single cell. (H) Oscillation amplitude for changes in cell area in embryos injected with water (blue) or 100 mM Y-27632 (red).
*Figure 1 continued on next page*

*Figure 1 continued*

Asterisks indicate p < 0.001. (I–J') Vertex resolution during axis elongation in embryos expressing E-cadherin:GFP and injected with water (I) or with 100 mM Y-27632 (J, J'). Arrowheads indicate nascent DV interfaces. (K) Distribution of vertex resolution angles relative to the AP axis in embryos injected with water (blue, *n* = 28 vertices in 3 embryos) or 100 mM Y-27632 (red, *n* = 25 interfaces in 3 embryos). Angles were measured 150 s after the onset of vertex resolution. An angle of 90° with respect to the AP axis corresponds to the DV axis. (L) Length of new DV interfaces forming within 30° of the AP axis in embryos injected with water (blue, *n* = 25 interfaces in 3 embryos) or 100 mM Y-27632 (red, *n* = 11 interfaces in 3 embryos). (A, I–J') Anterior left, dorsal up. Scale bars, 5 µm. (B, F, G, L) Time is with respect to the onset of vertex resolution, defined as the first time point in which the length of the nascent interface exceeded 1 µm. (H, K, L) Error bars, s.e.m. AP, anterior-posterior; DV, dorsal-ventral.

The following figure supplements are available for figure 1:

**Figure supplement 1.** Axis elongation in *Drosophila* is driven by neighbour exchange events.

**Figure supplement 2.** Dorsal and ventral cells oscillate with new DV interfaces.

**Figure supplement 3.** Directional assembly of new DV interfaces during vertex resolution requires actomyosin contractility.

**Figure supplement 4.** Par complex localization is affected by Y-27632, but not by Cytochalasin D.

**Figure supplement 5.** Oriented assembly of new DV interfaces requires actin-based contraction.

30° of the DV axis in Y-27632-injected embryos, in contrast to 0/28 in water-injected controls (p = 0.02, *Figure 1I,J',K*, *Video 2*). In addition, for vertices that resolved along the AP axis, inhibiting Rho-kinase reduced the rate of new edge elongation with respect to controls (0.001 ± 0.080 µm/min vs. 0.28 ± 0.06 µm/min, respectively, p = 0.01, *Figure 1I,J,L*, *Video 2*), suggesting that myosin activity facilitates the assembly of new DV interfaces. Similar results were obtained in embryos expressing Resille:GFP, a different cell outline marker (*Figure 1—figure supplement 3B–E*). However, Rho-kinase activity can regulate the localization of the Par polarity complex (*Atwood and Prehoda, 2009*; *Simões et al., 2010*) (*Figure 1—figure supplement 4A*), raising the possibility that abnormal vertex resolution upon Y-27632 injection was a consequence of defects in cell polarity, rather than reduced actomyosin contractility.

To further investigate the role of mechanical forces in vertex resolution, we disrupted actomyosin contractility by injecting embryos with 5 mM of Cytochalasin D, a drug that blocks actin polymerization by binding to the elongating end of filaments and preventing the addition of new actin monomers (*Flanagan and Lin, 1980*). Cytochalasin D injection disrupted the actin cytoskeleton (*Figure 1—figure supplement 5A–B*) and reduced apical area oscillations (p = 0.04, *Figure 1—figure supplement 5C–E*), without affecting the localization of Par-6, a member of the Par complex (*Figure 1—figure supplement 4B*). Cytochalasin D treatment led to an 83% reduction in the rate of new DV edge assembly with respect to controls (0.07 ± 0.10 µm/min vs. 0.40 ± 0.05 µm/min, respectively, p = 0.01, *Figure 1—figure supplement 5F–G,J*, *Video 3*). In Cytochalasin D-injected embryos 4/15 vertices resolved along the DV axis, in contrast to 0/50 in DMSO-injected controls (p = 4.0

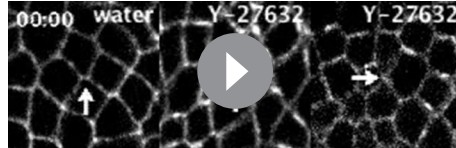

**Video 2.** Actomyosin contractility is required for directional vertex resolution. Germband cells expressing E-cadherin:GFP in embryos injected with water (left) or 100 mM Y-27632 (centre and right). A stack was acquired every 10 s. Time is indicated as min: s. Anterior left, dorsal up. This video relates to *Figure 1*.

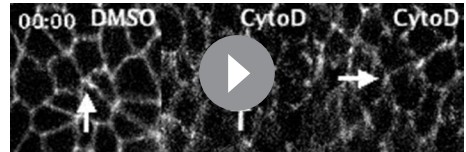

**Video 3.** Stabilization of actin filaments impairs directional vertex resolution. Germband cells expressing E-cadherin:GFP in embryos injected with 50% DMSO (left) or 5 mM Cytochalasin D (centre and right). A stack was acquired every 10 s. Time is indicated as min:s. Anterior left, dorsal up. This video relates to *Figure 1—figure supplement 5*.

$\times 10^{-15}$, *Figure 1—figure supplement 5F,H–I*, *Video 3*). Strikingly, in Cytochalasin D-injected embryos, 32/47 vertices persisted for at least 10 min and up to 40 min (*Figure 1—figure supplement 5K*). Together, our results demonstrate that actomyosin contractility is necessary for the directional assembly of new interfaces during vertex resolution in *Drosophila* axis elongation.

If actomyosin contractility in the cells anterior and posterior to a resolving vertex drives directional interface assembly, then the nascent edge must be under tension. To quantify tension, we used an ultraviolet (UV) laser to locally irradiate and sever DV interfaces in embryos expressing E-cadherin:GFP, and particle-tracking velocimetry to quantify the change in position of the tricellular vertices once connected by the severed interface. The instantaneous retraction velocity of the vertices is proportional to the tension sustained by the interface prior to ablation (*Hutson et al., 2003*; *Fernandez-Gonzalez et al., 2009*). We compared retraction velocities after ablation of control DV junctions that were not actively elongating (average length of $7.3 \pm 0.3$ μm, *Figure 2A*) and newly forming DV edges (average length of $3.4 \pm 0.2$ μm, *Figure 2B*). The retraction velocity after ablation of new DV junctions was $0.81 \pm 0.08$ μm/s, 32% greater than the retraction velocity after severing control DV edges ($0.61 \pm 0.03$ μm/s, p = 0.05, *Figure 2C*), indicating that – assuming uniform viscoelastic properties – new DV edges sustain increased mechanical tension with respect to non-elongating edges with similar orientation. New DV interfaces displayed smaller angles between the anterior or posterior cell junctions ($\theta_{avg} = 136.6 \pm 3.3°$) than control DV interfaces, ($\theta_{avg} = 150.3 \pm 3.4°$, p = 0.02, *Figure 2A–B,D*), and the retraction velocity after laser ablation was significantly anti-correlated with the angle between the anterior or posterior cell junctions ($r = -0.6$, p = $2.9 \times 10^{-5}$). Notably, no correlation was found between control or new DV interface length and instantaneous retraction velocity after ablation ($r = 0.04$ and $0.35$, respectively, *Figure 2E,F* and *Figure 2—figure supplement 1*), suggesting that differences in retraction velocity between control and new DV edges are independent from interface length, and determined by whether the edge is being assembled. Vertex retraction after laser ablation could result from actomyosin contractility at the interface or at another structure (for example, another interface or a medial apical surface) connected to the severed interface. New DV edges were myosin-depleted (*Blankenship et al., 2006*) (p = $4.3 \times 10^{-5}$, *Figure 2—figure supplement 2*), suggesting that vertex retraction after ablation of new DV edges was caused by tension generated elsewhere and exerted onto the new edge. Together, our data strongly suggest that mechanical tension parallel to the AP axis of the embryo contributes to vertex resolution.

To further investigate the relative contribution of anterior/posterior and dorsal/ventral cells to new DV junction assembly during vertex resolution, we disrupted actomyosin contractility specifically in the anterior and posterior, or the dorsal and ventral cells. To this end, we used a UV laser to irradiate and destroy myosin networks in the cells anterior/posterior or dorsal/ventral to four-cell vertices. Cells expressed E-cadherin:GFP to visualize cell outlines, and myosin:mCherry to track the assembly of contractile networks. Cells were re-irradiated upon assembly of medial actomyosin networks to prevent the generation of contractile forces. Irradiated cells were not extruded in the course of these experiments. Controls were four-cell vertices in which the anterior/posterior or dorsal/ventral cell pairs were sham-irradiated with the UV laser fully attenuated using a neutral density filter. When the contractile activity of anterior/posterior cells was disrupted, 4/7 four-cell vertices did not resolve (their length was never greater than 1 μm for at least 1 min), in contrast to 0/10 vertices in sham-irradiated controls. In controls, the rate of new edge elongation calculated over 180 s was $0.47 \pm 0.08$ μm/min (*Figure 3A,C*). Preventing contraction of the anterior/posterior cells resulted in a significant reduction of the rate of new edge elongation to $0.18 \pm 0.05$ μm/min for the vertices that resolved (p = 0.03; *Figure 3B–C*). These results suggest that contractility in the cells anterior and posterior to a multicellular vertex is necessary for vertex resolution and the assembly of the new DV interface.

To investigate the role of dorsal/ventral cells in vertex resolution, we prevented assembly and contraction of medial actomyosin networks in the dorsal and ventral cells using laser ablation. In contrast with the ablation of anterior/posterior cells, ablation of the DV cells did not prevent vertex resolution: 5/7 new DV interfaces reached a length of at least 1 μm, similar to 10/10 in controls. The initial rates of elongation were similar, with new DV interfaces elongating at a rate of $0.37 \pm 0.11$ μm/min over 60 s when contraction of the DV cells was disrupted, compared to rates of $0.50 \pm 0.15$ μm/min in sham-irradiated controls (p = 0.48, *Figure 3D–F*). However, ablation of the dorsal and ventral cells resulted in a significant reduction of the rate of new interface elongation over the subsequent 120 s of elongation, from $0.52 \pm 0.11$ μm/min in controls to $-0.03 \pm 0.08$ μm/min (p = 0.01, *Figure 3D–F*). Notably, in 3/5 vertices that resolved when dorsal/ventral cells were ablated, new DV

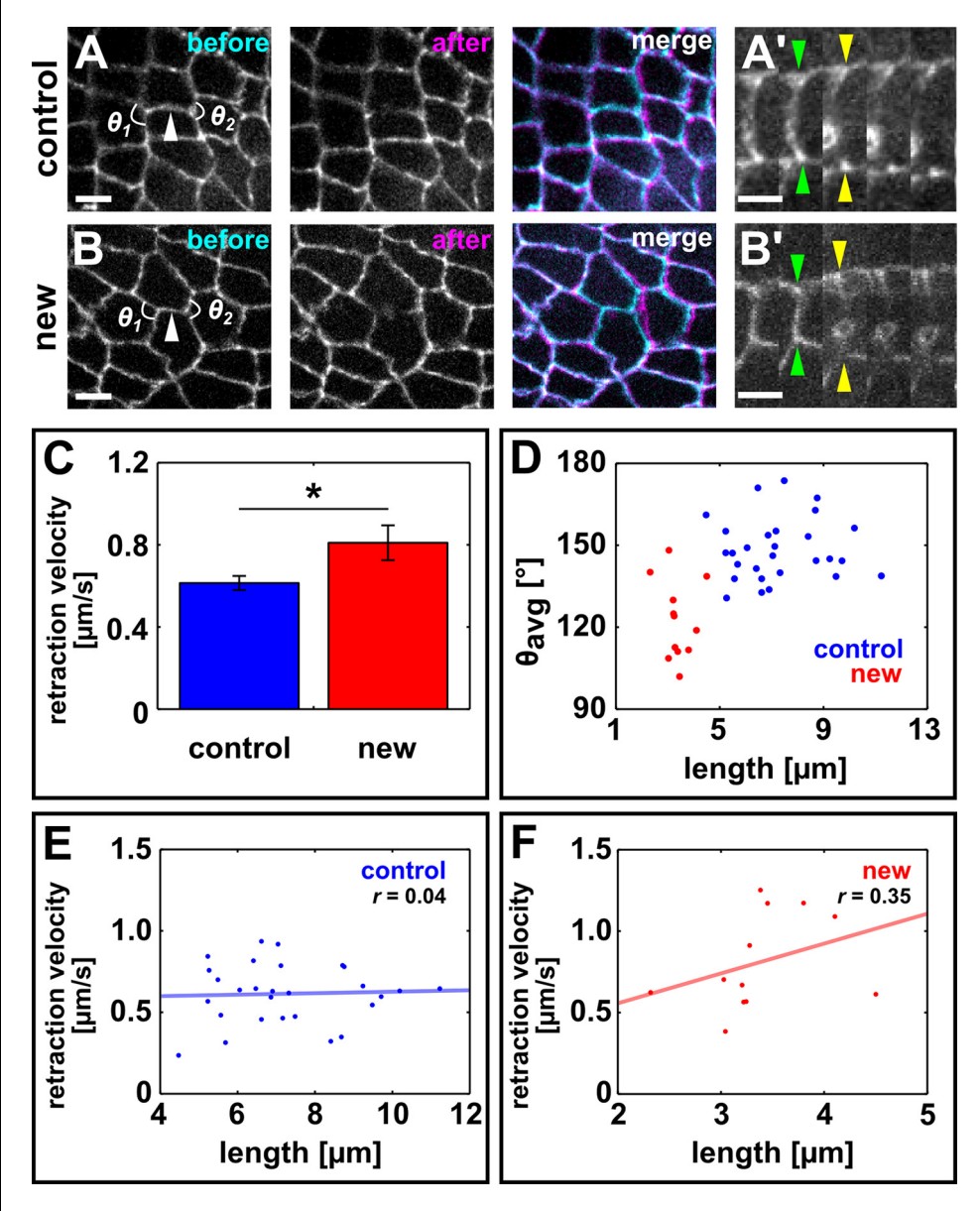

**Figure 2.** Resolving edges sustain increased mechanical tension during axis elongation. (**A, B**) Germband cells expressing E-cadherin:GFP before and after ablation of a control DV edge (**A**) or a newly forming DV edge (**B**). White arrowheads point to the ablated interface. $\theta_1$ and $\theta_2$ indicate the angles between the junctions anterior and posterior to the ablated interface, respectively. Anterior left, dorsal up. Scale bars, 5 µm. (**A', B'**) Kymographs showing the vertex displacement caused by laser ablation of the edges shown in (**A, B**). Arrowheads indicate vertex position prior to ablation (green) or immediately after (yellow). Interfaces are rotated by 90° with respect to (**A, B**) Anterior down, dorsal left. Scale bar, 3 s. (**C**) Retraction velocity after laser ablation in control (blue, $n = 28$) and new (red, $n = 12$) DV interfaces. Asterisk indicates p = 0.05. Error bars, s.e.m. (**D**) Scatterplot showing interface length vs. average junction angle at the anterior and posterior ends ($\theta_{avg} = (\theta_1 + \theta_2)/2$). (**E, F**) Scatterplots showing interface length vs. retraction velocity after laser ablation for control (**E**) and new (**F**) DV interfaces. Solid lines are best-fit lines. DV, dorsal-ventral.

The following figure supplements are available for figure 2:

**Figure supplement 1.** The retraction velocity after ablation of new and control DV edges is not anti-correlated with their length.

*Figure 2 continued on next page*

*Figure 2 continued*

**Figure supplement 2.** New DV edges do not display a significant myosin accumulation.

edges formed but were not sustained beyond 1 min, collapsing back into vertices. Together, our data suggest that dorsal/ventral cells are necessary to sustain the elongation of new DV interfaces, but not the resolution of multicellular vertices.

To determine if mechanical tension from the anterior and posterior cells can promote the elongation of new DV interfaces during germband extension, we developed a method to apply ectopic local tension to resolving vertices based on wound healing (*Campinho et al., 2013*; *Fernandez-Gonzalez and Zallen, 2013*). Upon wounding by irradiation with a UV laser, germband cells undergo apical constriction driven by medial-apical actomyosin networks (*Figure 4—figure supplement 1A*). Apical constriction of germband cells generates ectopic tension on the surrounding cell interfaces (*Figure 4—figure supplement 1A*, arrowheads). We used a UV laser to wound the cells anterior and posterior to resolving vertices by irradiating their medial-apical surfaces (*Figure 4A–B* and *Figure 4—figure supplement 1B*, and *Video 4*). Under sham-irradiation (UV laser fully attenuated using a neutral density filter), the cell area and medial myosin of the anterior and posterior cells remained largely unaffected, and the new DV interface elongated at a rate of 0.79 ± 0.14 µm/min (*Figure 4A, C,E*). Conversely, when the cells anterior and posterior were irradiated with UV light, myosin accumulated on the apical surface of the wounded cells and their apical areas decreased rapidly (*Figure 4B,D*), resulting in ectopic, AP-oriented tension on the resolving vertex. Under ectopic

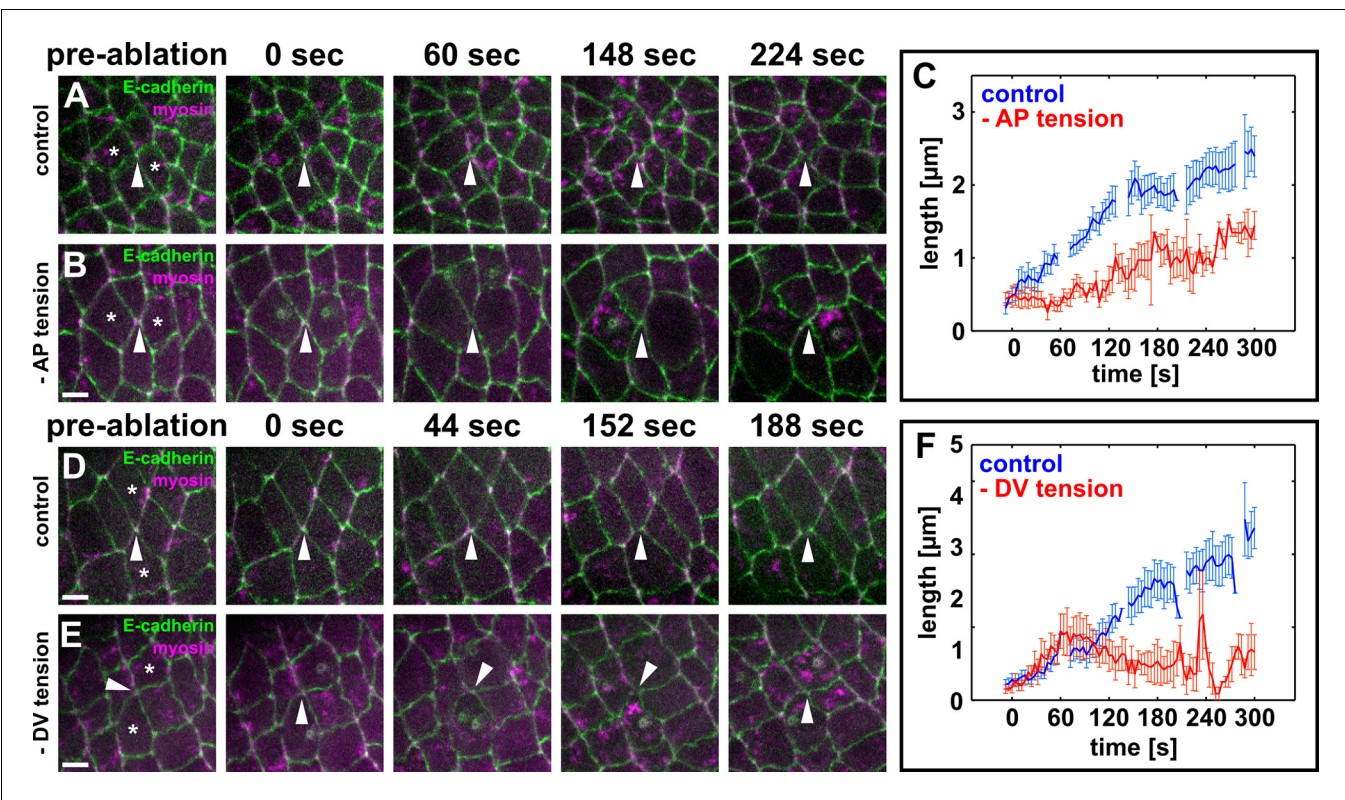

**Figure 3.** Local actomyosin contractility is necessary for vertex resolution and new DV interface assembly. (A, B, D, E) Cells expressing E-cadherin:GFP (green) and myosin:mCherry (magenta) in sham-irradiated controls (A, D) or when UV irradiation was used to reduce local tension (B, E). White arrowheads indicate resolving interfaces. Asterisks show the targeted cells. Time is with respect to the first laser irradiation. Anterior left, dorsal up. Scale bars, 5 µm. (C, F) Length of resolving DV interfaces over time in controls (blue, *n* = 10 interfaces in C and F), under reduced AP tension (red, *n* = 7 interfaces in C), or under reduced DV tension (red, *n* = 7 interfaces in F). Discontinuities in the blue lines indicate times at which cells were targeted with the attenuated UV laser in all experiments. Error bars, s.e.m. DV, dorsal-ventral.

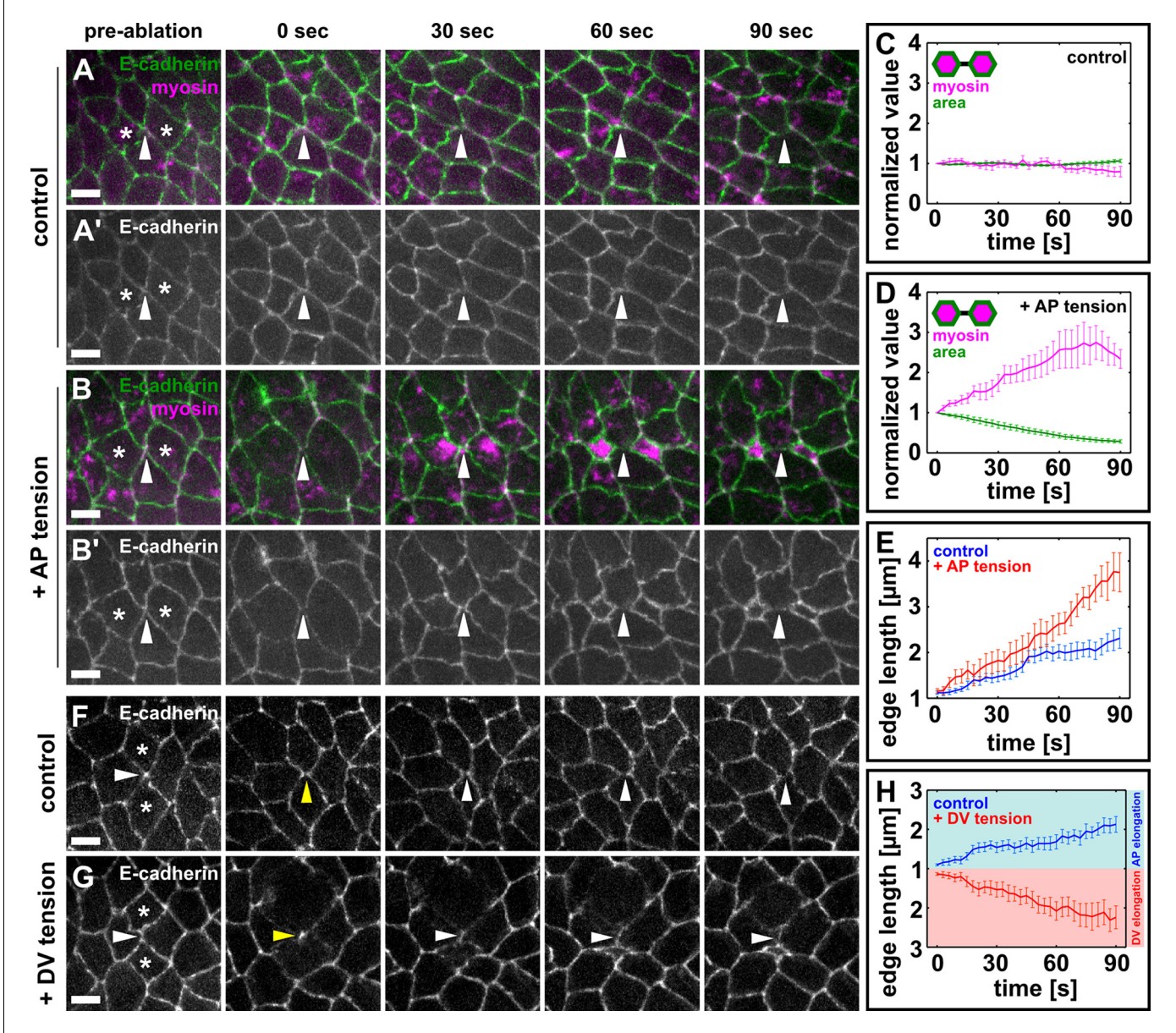

**Figure 4.** Local mechanical tension is sufficient to promote and orient new interface assembly during vertex resolution. (A–B') Cells expressing E-cadherin:GFP (green) and myosin:mCherry (magenta) in sham (A) or UV-irradiated (B) embryos. (C, D) Medial myosin intensity (magenta) and cell area (green) in sham (C, $n$ = 22 cells in 11 embryos) and UV-irradiated embryos (D, $n$ = 16 cells in 8 embryos). (E) Length of resolving DV interfaces over time in controls (blue, $n$ = 11 interfaces) and under increased tension along the AP axis (red, $n$ = 8 interfaces). (F, G) Cells expressing E-cadherin:GFP in sham (F) or UV-irradiated (G) embryos. Asterisks show the cells around a four-cell vertex (white arrowheads) that were irradiated. Yellow arrowheads indicate the formation of a four-cell vertex. (A, B, F, G) Anterior left, dorsal up. Scale bars, 5 μm. (H) Length of resolving interfaces over time in controls (blue, $n$ = 12) and under increased tension along the DV axis (red, $n$ = 13). Turquoise indicates elongation parallel to the AP axis, pink denotes DV elongation. (C–E, H) Time is with respect to the time point when the nascent DV interface first exceeded 1 μm in length. Error bars, s.e.m. (C, D) Normalization is with respect to the value at 0 s. AP, anterior-posterior; DV, dorsal-ventral.

The following figure supplement is available for figure 4:

**Figure supplement 1.** Wounded cells undergo apical constriction and induce ectopic tension on adjacent cell-cell junctions.

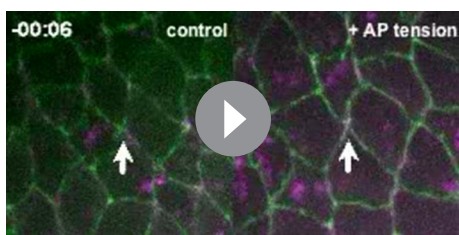

**Video 4.** Mechanical tension promotes rapid elongation of new DV interfaces. Germband cells in embryos expressing E-cadherin:GFP (green) and myosin:mCherry (magenta) under sham irradiation (left) or upon wounding and apical constriction of the cells anterior and posterior to a multicellular vertex (right). Arrows indicate resolving multicellular vertices. A stack was acquired every 3 s. Time is indicated as min:s. Anterior left, dorsal up. This video relates to *Figure 4*. DV, dorsal-ventral.

tension parallel to the AP axis, new DV junctions elongated at a rate of 1.73 ± 0.30 µm/min, 2.1-fold faster than the elongation rate in controls (p = 6.2 × 10$^{-3}$, *Figure 4E*). These results indicate that local mechanical tension parallel to the AP axis is sufficient to promote rapid assembly of new DV interfaces during vertex resolution in germband extension.

Our findings that ectopic tension can increase the rate of new edge elongation suggest that tension parallel to the DV axis may change the direction of vertex resolution. We compared the orientation and rate of new edge elongation in sham-irradiated embryos (*Figure 4F*) and in embryos in which we induced apical constriction of the cells dorsal and ventral to a four-cell vertex, increasing tension along the DV axis (*Figure 4G*). All the four-cell vertices examined in control embryos (n = 12) resolved within 30° of the AP axis, and the rate of new interface assembly was 0.69 ± 0.13 µm/min (*Figure 4F,H*). When we applied ectopic tension along the DV axis, the rate of new edge elongation was not affected (0.74 ± 0.18 µm/min, p = 0.8), but the orientation of the new edge changed and occurred within 30° of the DV axis in 13 out of 13 cases (*Figure 4G,H*). Together, our data indicate that local tension can promote and orient the assembly of new cell-cell interfaces, suggesting a central role for mechanical forces during vertex resolution in *Drosophila* axis elongation.

## Discussion

Polarized junction remodelling drives changes in tissue architecture from worms to mice (*Walck-Shannon and Hardin, 2014*). While junctional contraction and disassembly in the context of cell intercalation have been extensively explored (*Bertet et al., 2004*; *Blankenship et al., 2006*; *Rauzi et al., 2008*; *Fernandez-Gonzalez et al., 2009*; *Levayer et al., 2011*; *Bosveld et al., 2012*; *Shindo and Wallingford, 2014*; *Lau et al., 2015*), little is known about the mechanisms that control the directional assembly of new cell contacts during neighbour exchange. We used quantitative imaging, and biophysical and pharmacological approaches to show that local mechanical forces can direct the assembly of new junctions during *Drosophila* germband extension. New junctions elongate in pulses anti-correlated with the periodic contractions of the cells anterior and posterior to the new contact. Inhibiting actomyosin contractility disrupts both the rate and directionality of new junction assembly. Disrupting contractility in the cells anterior and posterior to the new edge disrupts vertex resolution and slows down new edge elongation, while preventing contraction of the dorsal and ventral cells mainly affects the maintenance and lengthening of the new cell interface. Hypercontraction of the cells anterior and posterior to the new edge accelerates the rate of new edge assembly. Finally, applying ectopic tension orthogonal to the characteristic orientation of vertex resolution is sufficient to alter the direction of new edge formation, suggesting that mechanical forces associated with actomyosin contractility direct the assembly of new cell contacts during multicellular vertex resolution in germband extension.

We show that vertex resolution occurs under increased mechanical tension, in a process that requires actomyosin contractility. Consistent with this, expression of inactive or constitutively active forms of myosin in embryos lacking the wild-type motor protein disrupts the directionality of vertex resolution during germband extension (*Kasza et al., 2014*). Furthermore, mechanical tension is necessary for directional resolution of multicellular vertices in the mouse embryonic ectoderm during limb bud elongation (*Lau et al., 2015*). In the *Drosophila* dorsal thorax, whose architecture is determined by neighbour exchange events, actomyosin contractility in new edges is tightly regulated to facilitate their elongation (*Bardet et al., 2013*). Our data suggest that the increase in tension on the new contact may be caused locally by the pulsatile, anisotropic contraction of the cells around the

resolving vertex. Interestingly, cells in the mouse limb bud ectoderm also display pulsed contractions that are disrupted in β-catenin mutants, and in these mutants the directionality of vertex resolution is lost (*Lau et al., 2015*). Together, these results are consistent with a general role for pulsed contractile activity in orienting and promoting cell intercalation.

We find that anterior/posterior and dorsal/ventral cells may play different roles during multicellular vertex resolution. Our data suggest that the anterior and posterior cells contribute to both vertex resolution and new edge elongation, while the dorsal and ventral cells are mainly necessary to support the elongation of the edge once the vertex has resolved. Recent mathematical modelling predicts that periodic actomyosin contractility in the medial-apical surface of anterior and posterior cells could drive the assembly of new edges during germband extension (*Lan et al., 2015*). The pulsed contraction of the anterior and posterior cells could cause rapid membrane reorganization in the dorsal and ventral cells (*Pramanik et al., 2009*), facilitating the assembly of an actin scaffold (*Pickering et al., 2013*) and the formation of junctions. Junctional and cytoskeletal remodelling require intact DV cells, and possibly, the continued stimulus from AP cell pulsing. The implementation of optogenetic approaches (*Guglielmi et al., 2015*) to locally inhibit membrane remodelling and junctional and cytoskeletal dynamics will reveal how these processes are coordinated across cells to promote directional cell rearrangements during epithelial morphogenesis.

The mechanisms by which mechanical tension regulates the assembly of new cell interfaces during germband extension remain unclear. An accumulation of filamentous actin is the first known step of vertex resolution (*Blankenship et al., 2006*), and in this study, we found that blocking actin polymerization results in multicellular vertices that do not resolve. Thus, actin polymerization may play a central role in vertex resolution. Mechanical forces can control actin dynamics in vitro, possibly by inducing conformational changes in the formin family of actin regulators to favour faster and more frequent polymerization of actin filaments (*Courtemanche et al., 2013*; *Higashida et al., 2013*; *Jegou et al., 2013*). In addition, actin filaments are less susceptible to severing in the presence of increased tension (*Hayakawa et al., 2011*), which may accelerate actin assembly at nascent cell interfaces. Understanding how mechanical forces impact the localization and dynamics of different actin regulators will contribute to elucidating the mechanisms by which tension promotes directional cell behaviours during *Drosophila* axis elongation.

## Materials and methods

### Fly stocks

We used the following markers for live imaging: *ubi-E-cadherin:GFP* (*Oda and Tsukita, 2001*), *sqh-sqh:mCherry* (*Martin et al., 2009*), *resille:GFP* (*Morin et al., 2001*), *sqh-GFP:utrophin* (*Rauzi et al., 2010*), and *par-6^{Δ226}, par-6:GFP* (*Wirtz-Peitz et al., 2008*).

### Time-lapse imaging

Stage-7 embryos were dechorionated in 50% bleach for 90 s, rinsed, glued ventrolateral side down to a glass coverslip using heptane glue, and mounted in a 1:1 mix of halocarbon oil 27 and 700 (Sigma-Aldrich, St. Louis, MO). Embryos were imaged using a Revolution XD spinning disk confocal microscope equipped with an iXon Ultra 897 camera (Andor, Belfast, UK) and a 1.5x coupling lens. For experiments using laser ablation, a 60x oil immersion lens (Olympus, Shinjuku, Japan; NA 1.35) was used; for all other experiments, a 40x oil immersion lens (Olympus, NA 1.35) was used. Sixteen-bit Z-stacks were acquired at 0.3-μm steps every 3–10 s (8–10 slices per stack).

### Laser ablation

Ablations were induced using a pulsed Micropoint $N_2$ laser (Andor) tuned to 365 nm. The laser delivers 120 μJ pulses at durations of 2–6 ns each. For ablation of cell boundaries, 10 consecutive laser pulses were delivered to a single spot along a cell interface. For single-cell wounds, 10 consecutive laser pulses were delivered to each of two spots spaced 2 μm apart on the medial-apical region of the cell of interest. In experiments where local tension was reduced, 10 laser pulses were delivered to a single spot on the medial-apical region of the cell of interest. Cells were re-ablated upon assembly of medial-apical myosin networks. In sham-irradiated controls, cells were targeted with the laser completely attenuated every 60 s to mimic the repeated ablations performed in the corresponding experiments.

## Drug injections

Embryos were dechorionated and glued to a coverslip as above, dehydrated for 10–15 min, and covered with a 1:1 mix of halocarbon oil 27 and 700 (Sigma-Aldrich). Embryos were injected using a Transferman NK2 micromanipulator (Eppendorf, Hamburg, Germany), and a PV820 microinjector (WPI, Sarasota, FL) attached to the spinning disk confocal microscope. Drugs (Y-27632, Tocris Bioscience, Bristol, UK); (Cytochalasin D, EMD Millipore, Darmstadt, Germany) were injected into the perivitelline space, where they are predicted to be diluted 50-fold (*Foe and Alberts, 1983*). Y-27632 was injected at 100 mM in water; control embryos were injected with water. Cytochalasin D was injected at 5 mM in 50% DMSO; control embryos were injected with 50% DMSO. Embryos were imaged immediately after injection for at least 10 min.

## Cell segmentation, tracking, and quantification

Image analysis was performed using algorithms developed with Matlab (MathWorks, Natick, MA) and DIPImage (Delft University of Technology, Delft, Netherlands) and integrated in our custom Scientific Image Segmentation and Analysis (SIESTA) software (*Fernandez-Gonzalez and Zallen, 2011*; *Leung and Fernandez-Gonzalez, 2015*).

The onset of vertex resolution was established as the first time at which the length of a nascent interface exceeded 1 μm. New edge orientation was quantified relative to the AP axis of the embryo, defined as 0°, and was measured 150 s after the onset of vertex resolution. Edge length was measured as the distance between the two vertices defining the edge. To measure how fast new edges assemble, we defined the rate of elongation at time $t$ as:

$$\text{rate of elongation } (t) = \frac{l(t) - l(t_0)}{t - t_0} \tag{1}$$

where $l(t)$ represents the length of the edge at time $t$, and $t_0$ is the time of onset of vertex resolution. The rate of elongation was calculated over the initial 90 s of interface elongation, unless indicated otherwise. Cell areas were quantified using an algorithm in which seeds were manually placed within each cell of interest in the first timepoint of a movie. Seeds were automatically expanded to delineate the cell boundaries using the watershed method (*Beucher, 1992*), a region-growing algorithm. Seeds were subsequently propagated to the next time point using particle image velocimetry to account for cellular movement (Wang and Fernandez-Gonzalez, in preparation), and the process was iterated. To measure retraction velocities following laser ablation, we determined the change in distance between the two vertices delimiting the ablated interface, and divided this value by the sum of the ablation and the stack acquisition times.

In time-lapse images, fluorescence was measured from maximum intensity projections of three apical slices. Fluorescence intensities were background-subtracted using the most frequent pixel value (the mode) of a maximum intensity projection of three basal slices cropped around the region of interest (10 μm × 10 μm). Intensity values were corrected for photobleaching by dividing by the mean image intensity in each time point. To quantify myosin levels in new DV edges with respect to AP edges, we imaged embryos expressing myosin:mCherry, and measured fluorescence in manually traced cell interfaces. We subtracted the image mode from the myosin fluorescence measurements as an estimate of the background.

Oscillatory cell behaviours were characterized by the rate of change per minute of the corresponding magnitude, calculated as the difference of measurements collected 1 min apart. To calculate periods, rates of change were detrended by subtracting the line of best fit using the *detrend* function in Matlab (Mathworks). The period was computed as the inverse of the dominant frequency in a fast Fourier transform of the detrended signal. To calculate the mean change in edge length during the elongation or shortening steps of new DV edge formation, we quantified the area under the curve for positive (elongation) or negative (shortening) rates of length change. The resulting numbers were the total elongation or shortening for a given edge, which divided by the number of pulses yielded the mean change in length per elongation or shortening pulse. The correlation between signal pairs was determined using the *corrcoef* function in Matlab (Mathworks). To find the time shift required for minimum or maximum correlation between signal pairs, one signal was shifted forward and backward in time relative to the other, in increments of 10 s up to 240 s. With each increment, the correlation was recalculated. The resulting correlation curve was Gaussian-smoothed

using a sigma of 10 s, and the time shifts required to obtain the first local minimum and maximum in the correlation values were determined.

## Statistical analysis

Sample means were compared using Student's *t*-test (*Glantz, 2002*). The significance of correlation coefficients was calculated by transforming the correlation value into a *t*-statistic using the Matlab *corrcoef* function (Mathworks). Sample distributions were contrasted using Kolmogorov–Smirnov's test. Error bars indicate the standard error of the mean (s.e.m.).

## Acknowledgements

We thank Ashley Bruce, Sevan Hopyan, Jennifer Zallen, Miranda Hunter, Anna Kobb, Michael Wang, and Teresa Zulueta-Coarasa for comments on the manuscript, and Tony Harris for reagents. This work was supported by an Ontario Early Researcher Award to RFG, and grants from the University of Toronto Faculty of Medicine Dean's New Staff Fund, the Canada Foundation for Innovation (#30279), and the Natural Sciences and Engineering Research Council of Canada Discovery Grant program (#418438-13) to RFG. JCY was partially supported by a Barbara and Frank Milligan Graduate Fellowship and a Hayden Hantho Award (Ontario Student Opportunity Trust Fund). Flybase provided important information for this study. The authors declare no competing financial interests.

## Additional information

### Funding

| Funder | Grant reference number | Author |
| --- | --- | --- |
| Natural Sciences and Engineering Research Council of Canada | Discovery Grant, 418438-13 | Rodrigo Fernandez-Gonzalez |
| Canada Foundation for Innovation | Leaders Opportunity Fund, 30279 | Rodrigo Fernandez-Gonzalez |
| Ontario Ministry of Economic Development and Innovation | Research Fund, 30279 | Rodrigo Fernandez-Gonzalez |
| University of Toronto | Faculty of Medicine Dean's New Staff Fund, DF-2012-02 | Rodrigo Fernandez-Gonzalez |
| Ontario Ministry of Economic Development and Innovation | Early Researcher Award, ER14-10-170 | Rodrigo Fernandez-Gonzalez |
| University of Toronto | Barbara and Frank Milligan Graduate Fellowship | Jessica C Yu |
| University of Toronto | Hayden Hantho Award | Jessica C Yu |

The funders had no role in study design, data collection and interpretation, or the decision to submit the work for publication.

### Author contributions

JCY, Conception and design, Acquisition of data, Analysis and interpretation of data, Drafting or revising the article; RFG, Conception and design, Analysis and interpretation of data, Drafting or revising the article

### Author ORCIDs

Rodrigo Fernandez-Gonzalez, http://orcid.org/0000-0003-0770-744X

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
