## [Decision Letter]

Thank you for submitting your work entitled "Local mechanical forces promote polarized junctional assembly and axis elongation in *Drosophila*" for consideration by *eLife*. Your article has been reviewed by two peer reviewers, and the evaluation has been overseen by John Wallingford (Reviewing Editor) and Randy Schekman (Senior Editor).

The reviewers have discussed the reviews with one another and the Reviewing editor has drafted this decision to help you prepare a revised submission.

This manuscript analyses the mechanisms of junction elongation during convergent extension in the *Drosophila* germband embryo. The manuscript is of general interest since cell rearrangements are fundamental to many morphogenetic processes in both invertebrates and vertebrates. Such rearrangements involve shrinkage and expansion of junctions, and while most work in this area has focused on junction shrinkage, this work is notable for its focus on junction elongation. The authors propose that local mechanical forces due to neighbor cell apical constrictions promote junction elongation and influence their direction of elongation.

The consensus of the reviewers is that some additional work is needed to make the paper suitable for *eLife*.

Major comments:

1) The reviewers both find the correlation between DV junction elongation and the contractions of the neighboring cells to be of interest, but not entirely convincing as shown. The elongation of DV junctions was shown by the authors to be anti-correlated with changes of the surface areas of the Anterior and Posterior cells, but at the same time to be correlated with the surface area changes of Dorsal and Ventral cells. The question arises, then, whether the receding of the junction is due to loss of tension in the A/P cells, or rather due to contraction of the D/V cells? We ask the authors to clarify this situation. One possibility is asking if laser cutting of the anterior and posterior cells, perpendicular to the DV junction, releases the tension along the junction and thus prevents elongation. A related question is whether these AP contractions contribute only to junction elongation or also to the initial resolution of the multicellular vertices.

2) A second question relates to tension in elongating DV junctions across their lifetime. In the Results section the authors state that "If actomyosin contractility in the cells anterior and posterior to the resolving vertex drives directional vertex resolution, then the nascent edge must be under tension". However, force balance is sufficient to generate a tensed junction upon formation, and it is well known that DV edges prior to cell rearrangement are characterized by a lower tension (they do not accumulate Myosin II). Therefore, the tension of the newly formed edge should be compared to the average junction tension. Overall it is difficult to be sure that the difference observed between "new" junctions and "control" junctions is not just due to their average difference in length (Figure 2). The authors need to compare the tension in a pool of "old" and "new" DV junctions which have the same length averages. It would also be of interest to compare retraction velocities of freshly formed DV junctions in different times of the elongation cycle more generally, i.e. is the retraction speed different in shortening vs. expanding junctions?

3) Atwood and Prehoda (CB, 2009) have shown that Y-27632 inhibits aPKC. The specificity of the Y-27632 treatment should therefore be established under the authors' experimental conditions. Is aPKC lost from the cortex in these animals?

[Editors' note: further revisions were requested prior to acceptance, as described below.]

Thank you for resubmitting your work entitled "Local mechanical forces promote polarized junctional assembly and axis elongation in *Drosophila*" for further consideration at *eLife*. Your revised article has been favorably evaluated by Randy Schekman (Senior editor) and a Reviewing editor. The manuscript has been significantly improved by the new data added to the revision. However, we must still ask that you make one final modification to improve your manuscript before acceptance.

Our concern about the Y-27632 was quite specific, and while the new experiments included here do allay that concern, their presentation in the current form of the paper obscures the key issue. In Figure 6, the Y compound had a profound effect on Par localization, exactly what we feared. Nonetheless, because cytochalasin had no such effect, your overall arguments remain convincing. Ideally, these data on Par complex localization should be discussed, as they do provide some important insights. If there's a good reason to leave the data out, this may be acceptable; but at a minimum, the paper needs to explicitly acknowledge that the Y compound has pleiotropic effects here, while the cytochalasin data make a strong case for actomyosin explaining the phenotypes reported.

---

## [Author Response]

*Major comments: 1) The reviewers both find the correlation between DV junction elongation and the contractions of the neighboring cells to be of interest, but not entirely convincing as shown. The elongation of DV junctions was shown by the authors to be anti-correlated with changes of the surface areas of the Anterior and Posterior cells, but at the same time to be correlated with the surface area changes of Dorsal and Ventral cells. The question arises, then, whether the receding of the junction is due to loss of tension in the A/P cells, or rather due to contraction of the D/V cells? We ask the authors to clarify this situation. One possibility is asking if laser cutting of the anterior and posterior cells, perpendicular to the DV junction, releases the tension along the junction and thus prevents elongation. A related question is whether these AP contractions contribute only to junction elongation or also to the initial resolution of the multicellular vertices.* To determine the relative contribution of anterior/posterior (A/P) or dorsal/ventral (D/V) cells to the elongation of new junctions parallel to the AP axis of the embryo, we ablated medial myosin networks in the cells A/P or D/V to resolving four-cell vertices. Repeated ablations were necessary to prevent the assembly of contractile actomyosin networks that drive pulsatile cell behaviours. We found that both A/P and D/V cells were necessary for the assembly of new DV interfaces (new Figure 3). Our data suggest that A/P and D/V cells may play different roles in vertex resolution and new interface elongation. When contraction of A/P cells was disrupted, vertex resolution was delayed with respect to controls, and the subsequent rate of new edge elongation decreased. In contrast, for D/V cell ablations, the initial steps of vertex resolution occurred at a similar rate as in controls, but after the new interface reached a length of approximately 1 µm, the elongation of the new interface stopped. We propose that contraction of A/P cells is necessary for the initial resolution of the multicellular vertex and the subsequent elongation of the new DV interface. D/V cells, on the other hand, may be necessary to sustain the elongation of the new DV interface that they share, by providing actin filaments and junctional molecules that stabilize that interface.

We have added a new figure (new Figure 3) demonstrating the effects of A/P and D/V ablations on vertex resolution. We have also described our new results in the text (Results section):

“To further investigate the relative contribution of anterior/posterior and dorsal/ventral cells to new DV junction assembly during vertex resolution, we disrupted actomyosin contractility specifically in the anterior and posterior, or the dorsal and ventral cells. […] Together, our data suggest that dorsal/ventral cells are necessary to sustain the elongation of new DV interfaces, but not the resolution of multicellular vertices.”

Finally, we have discussed the possibility that A/P and D/V cells have different contributions to vertex resolution and new junction assembly (Discussion):

“We find that anterior/posterior and dorsal/ventral cells may play different roles during multicellular vertex resolution. […] The implementation of optogenetic approaches (Guglielmi, 2015) to locally inhibit membrane remodelling and junctional and cytoskeletal dynamics in the cells adjacent to resolving vertices will reveal the relative contributions of anterior/posterior and dorsal/ventral cells during vertex resolution and new interface elongation.”

*2) A second question relates to tension in elongating DV junctions across their lifetime. In the Results section the authors state that "If actomyosin contractility in the cells anterior and posterior to the resolving vertex drives directional vertex resolution, then the nascent edge must be under tension". However, force balance is sufficient to generate a tensed junction upon formation, and it is well known that DV edges prior to cell rearrangement are characterized by a lower tension (they do not accumulate Myosin II). Therefore, the tension of the newly formed edge should be compared to the average junction tension. Overall it is difficult to be sure that the difference observed between "new" junctions and "control" junctions is not just due to their average difference in length (Figure 2). The authors need to compare the tension in a pool of "old" and "new" DV junctions which have the same length averages. It would also be of interest to compare retraction velocities of freshly formed DV junctions in different times of the elongation cycle more generally, i.e. is the retraction speed different in shortening vs. expanding junctions?*

Others and we showed that DV-oriented edges during germband extension are depleted of myosin (Bertet et al., 2004; Blankenship et al., 2006) and sustain reduced tension (Rauzi et al., 2008; Fernandez-Gonzalez et al., 2009). However, these studies did not distinguish between the more abundant, non-elongating, “control” DV junctions, and the less frequent “new” DV junctions. While new DV junctions also display low levels of myosin (Blankenship et al., 2006) and new Figure 2—figure supplement 2), our laser ablation data indicate that new DV junctions sustain increased tension with respect to control DV junctions. However, increased tension at new DV edges does not necessarily imply increased actomyosin contractility. Laser ablation experiments cannot distinguish between tension “actively” generated on a junction via actomyosin contractility at the junction itself, or tension “passively” sustained by a junction due to actomyosin contractility in other junctions or subcellular domains (e.g. the medial apical surface of cells) connected to the junction. In both situations the junction sustains increased tension, and ablation of the junction will result in the recoil of the vertices that the junction connected, at a velocity proportional to the tension sustained.

To clarify this point, we have added a new supplemental figure (new Figure 2—figure supplement 2) showing that new DV edges display relatively low myosin levels, suggesting that the tension that new DV edges sustain must be generated elsewhere. We have discussed this figure in the text (Results section):

“Vertex retraction after laser ablation could result from actomyosin contractility at the interface or at another structure (for example, another interface or a medial apical surface) connected to the severed interface. New DV edges were myosin-depleted (Blankenship et al., 2006) (*P* = 4.3x10_-5_, Figure 2—figure supplement 2), suggesting that vertex retraction after ablation of new DV edges was caused by tension generated elsewhere and exerted onto the new edge. Together, our data strongly suggest that mechanical tension parallel to the AP axis of the embryo contributes to vertex resolution.”

It is not possible to compare retraction velocities for control and new DV junctions of the same length, as by definition, new DV junctions will be shorter than control ones. However, if the difference in retraction velocity between new and control junctions was due to the reduced length of the new junctions, one would expect to see that shorter new junctions display greater recoil velocities than longer new junctions, and similarly for control junctions. Thus, to determine if the length of DV junctions correlates with the tension they sustain, we quantified the correlation between interface length and velocity of vertex retraction, both for control and new junctions. We found that there was no significant correlation between junction length and velocity of retraction after ablation (Figure 2), suggesting that the observed differences in recoil velocity between control and new junctions are not a consequence of the differences in length.

We have made the absence of a correlation between junction length and retraction velocity explicit, by including independent plots of interface length prior to ablation vs.retraction velocity (new Figure 2). On the new plots, we have indicated the value of the correlation between these two magnitudes to clearly demonstrate that interface length and ablation velocity are not correlated. This is also indicated in the text (Results section):

“Notably, no correlation was found between control or new DV interface length and instantaneous retraction velocity after ablation (*r* = 0.04 and 0.35, respectively, Figure 2 and Figure 2—figure supplement 1), suggesting that differences in retraction velocity between control and new DV edges are independent from interface length, and determined by whether the edge is being assembled.”

In addition to this, we have added a new Figure (Figure 2—figure supplement 1) showing examples of long edges with greater retraction velocities than short edges, both within the control and the new DV edge categories.

We investigated if tension on new DV junctions increased and decreased as junctions elongate or shrink, respectively. To this end, we conducted laser ablation experiments on new edges that were filmed for 2 min before ablation, and we calculated the rate of interface length change prior to ablation. Based on this, we classified interfaces as elongating, if their length was increasing immediately before ablation, or shortening, if their length was decreasing. The velocity of retraction for elongating interfaces (0.77 ± 0.10 µm/s, *n* = 10) was not significantly different from the velocity of retraction for shortening interfaces (0.74 ± 0.18 µm/s, *n* = 4, *P* = 0.87, Figure 5), suggesting that new DV interfaces do not experience significant changes in tension during the elongation and shortening phases of the edge assembly cycle. This result is consistent with the 6.8-fold greater magnitude of elongation pulses with respect to shortening pulses (772 ± 46 nm vs.114 ± 19 nm), which suggests that the increase in tension caused by the elongation phase may not be significantly alleviated by the shortening phase.

Author response image 1.New DV edges sustain similar tension during elongation and shortening.Retraction velocity after laser ablation of new DV edges at the shortening (blue, *n* = 4 interfaces in 4 embryos) and elongation (red, *n* = 10 interfaces in 10 embryos) phases of the new edge assembly cycle.**DOI:**
http://dx.doi.org/10.7554/eLife.10757.019

*3) Atwood and Prehoda (CB, 2009) have shown that Y-27632 inhibits aPKC. The specificity of the Y-27632 treatment should therefore be established under the authors' experimental conditions. Is aPKC lost from the cortex in these animals?*

To determine if Y-27632 affected the dynamics of aPKC during axis elongation, we used a genomic rescue line (*par-6_Δ226_, par-6:GFP* (Wirtz-Peitz et al., 2008)) to determine the localization of the Par polarity complex, of which aPKC is a member, in Y-27632-injected embryos and water-injected controls. This allowed us to investigate the localization of the Par complex using a reporter that does not cause overexpression phenotypes. Upon treatment with Y-27632, Par-6:GFP accumulated in “bars” at the centre of junctions (Figure 6), similar to previous reports for Par-3, another member of the Par complex (Simoes et al., 2010). Therefore, defects in vertex resolution observed in embryos injected with Y-27632 could be caused by the loss of myosin localization at the apical cell surface (Figure 1—figure supplement 3), by changes in the localization of the Par complex (Figure 6), or by both. To more specifically address the role of actomyosin contractility in the elongation of new DV junctions, we treated embryos with 5 mM of Cytochalasin D, a drug that inhibits actin polymerization (Flanagan and Lin, 1980) and has been used to disrupt actomyosin contractility in *Drosophila* embryos (Martin et al., 2009). We found that Par-complex localization was not disrupted by treatment with 5 mM Cytochalasin D (Figure 6). However, Cytochalasin D treatment disrupted both actin and myosin dynamics in germband cells (new Figure 1—figure supplement 4), and the oscillatory behaviours of germband cells were significantly attenuated (new Figure 1—figure supplement 4), suggesting that actomyosin contractility was disrupted. Similar to the results obtained when we injected embryos with Y-27632, we found that inhibition of actomyosin contractility using Cytochalasin D significantly affected the directionality and rate of new edge elongation during vertex resolution (new Figure 1—figure supplement 4), further suggesting that actomyosin contractility is necessary for vertex resolution during *Drosophila* axis elongation.

Author response image 2.Par complex localization is affected by Y-27632, but not by Cytochalasin D.(**A-B**) Germband cells expressing Par-6:GFP at endogenous levels and injected with water (**A**), 100 mM Y- 27632 in water (**A’**), 50% DMSO (**B**) or 5 mM Cytochalasin D in 50% DMSO (**B’**). Anterior left, dorsal up. Scale bars, 10 µm.**DOI:**
http://dx.doi.org/10.7554/eLife.10757.020

We have added a new figure demonstrating how actin and myosin localization change upon Cytochalasin D injection, and the effects on germband extension and vertex resolution of Cytochalasin D injection (new Figure 1—figure supplement 4). We have discussed these results in the text (Results section):

“We also disrupted actomyosin contractility by injecting embryos with 5 mM of Cytochalasin D, a drug that blocks actin polymerization by binding to the elongating end of filaments and preventing the addition of new actin monomers (Flanagan and Lin, 1980). […] Together, our results demonstrate that actomyosin contractility is necessary for the directional assembly of new interfaces during vertex resolution in *Drosophila* axis elongation.”

[Editors' note: further revisions were requested prior to acceptance, as described below.]

*Our concern about the Y-27632 was quite specific, and while the new experiments included here do allay that concern, their presentation in the current form of the paper obscures the key issue. In Figure 6, the Y compound had a profound effect on Par localization, exactly what we feared. Nonetheless, because cytochalasin had no such effect, your overall arguments remain convincing. Ideally, these data on Par complex localization should be discussed, as they do provide some important insights. If there's a good reason to leave the data out, this may be acceptable; but at a minimum, the paper needs to explicitly acknowledge that the Y compound has pleiotropic effects here, while the cytochalasin data make a strong case for actomyosin explaining the phenotypes reported.*

We have added a new figure to our manuscript (Figure 1—figure supplement 4), based on Figure 6. The new figure demonstrates the changes in localization of the Par complex upon Y-27632 treatment, and shows that there are no obvious defects in Par complex localization upon Cytochalasin D treatment. We have discussed these results in the text (Results section):

“However, Y-27632 treatment can affect the localization of the Par polarity complex (Atwood and Prehoda, 2009) (Figure 1—figure supplement 4), raising the possibility that abnormal vertex resolution upon Y-27632 injection was a consequence of defects in cell polarity, rather than reduced actomyosin contractility. […] Cytochalasin D injection disrupted the actin cytoskeleton (Figure 1—figure supplement 5) and reduced apical area oscillations (*P* = 0.04, Figure 1—figure supplement 5), without affecting the localization of the Par complex (Figure 1—figure supplement 4).”